# Validation of a High Flow Rate Puff Topography System Designed for Measurement of Sub-Ohm, Third Generation Electronic Nicotine Delivery Systems

**DOI:** 10.3390/ijerph19137989

**Published:** 2022-06-29

**Authors:** Evan Floyd, Toluwanimi Oni, Changjie Cai, Bilal Rehman, Jooyeon Hwang, Tyler Watson

**Affiliations:** 1Department of Occupational and Environmental Health, Hudson College of Public Health, University of Oklahoma Health Sciences Center, Oklahoma City, OK 73104, USA; toluwanimi-oni@ouhsc.edu (T.O.); changjie-cai@ouhsc.edu (C.C.); jooyeon-hwang@ouhsc.edu (J.H.); tyler@watson.do (T.W.); 2University of Oklahoma Health Sciences Center, University of Oklahoma College of Medicine, Oklahoma City, OK 73104, USA; bilal-rehman@ouhsc.edu

**Keywords:** electronic nicotine delivery systems, high flow rate puff topography, flow cell, tobacco regulatory control

## Abstract

There are few known puff topography devices designed solely for gathering electronic cigarette puff topography information, and none made for high-powered sub-ohm devices. Ten replicate Bernoulli flow cells were designed and 3D printed. The relationship between square root of pressure difference and flow rate was determined across 0–70 L/min. One representative flow cell was used to estimate puff volume and flow rate under six simulated puffing regimes (0.710 L, 2.000 L and 3.000 L, at low and high flow rates) to determine the system’s accuracy and utility of using dual pressure sensors for flow measurement. The relationship between flow rate and square root of pressure differential for the ten replicate cells was best fit with a quadratic model (R^2^ = 0.9991, *p* < 0.0001). The higher-pressure sensor was accurate at both low and high flow rates for 0.71 L (102% and 111% respectively), 2.00 L (96% and 103% respectively), and 3.00 L (100.1% and 107% respectively) but the lower-pressure sensor provided no utility, underpredicting volume and flow. This puff topography system generates very little resistance to flow, easily fits between user’s atomizer and mouthpiece, and is calibrated to measure flows up to 70 L/min.

## 1. Introduction

In recent times, there has been a rapid advancement in electronic cigarette (EC) technology, thereby resulting in the production of newer “high-powered” devices with advanced control capabilities. These newer, high-powered devices are largely referred to as third generation (3G) ECs [1,2,3,4,5]. There is a need to scientifically evaluate ECs [6] and most especially current generation devices for effects of variations in voltage and power settings [5,7], aerosol characteristics [5,8], heating coil types, and EC emissions [9] among other parameters. Variations in designs and the rapid metamorphosis of ECs may render previous studies focused on first and second generation (1G and 2G) ECs [10,11,12,13] obsolete with regards to the evaluation of safety, use, and performance of newer electronic cigarette products [14] that hinge on puff topography measurements [15,16,17].

The 1G EC is a cig-a-like, simple low-powered (3–8 W) device with rechargeable batteries and combined cartridges and atomizers [1,18,19,20] that does not have user adjustment and is operated simply by suction (puffing). On the other hand, 2G ECs are typically pen style devices with larger battery capacity and refillable reservoirs [1,19,20], user replaceable parts, variable voltage, and relatively low power (<20 W), and 3G ECs are essentially high powered (>20W) versions of 2G EC, typically with a boxy shape. Within the vaping community these are often called “mods” and “box-mods”. The 3G ECs are high powered and very customizable with replaceable heating coils and wicks for atomizers [1,20]. Some ECs have been referred to as 4G devices. However, 4G ECs are not well defined and the term is used inconsistently in literature. They have been described as sub-ohm devices having variable voltage capabilities and automatic temperature control for their mods [20,21] and as being next generation nicotine salt devices similar to JUUL vaping devices [22,23]. In the view of these authors, temperature control is a minor feature addition and still falls within 3G while low-powered, suction-activated devices such as JUUL and Puff Bar are merely effective retooling of a 1G platform, regardless of using nicotine salt or not.

Puff topography is the measurement or quantification of volume, duration, number, and flow rate of puffs as well as the intervals between puffs [24]. For ECs the topography should also include the specific operational conditions of the device since these are known to influence the aerosol properties [5,25,26] and nicotine yield [16,27,28] of ECs. To acquire a comprehensive understanding of EC puff topography, we must have topography devices capable and well-suited for measurement of the puff characteristics unique to each type of EC being used without altering the use behavior.

The way EC products are used is constantly evolving and diversifying. Accurate measurement of EC topography is essential in understanding use behaviors, nicotine delivery, and toxicant profiles. Real-world EC use behaviors must be determined for all types of EC products since each of the different generations are clearly used in different manners such as direct lung inhalation for 3G ECs versus mouth-to-lung puffing for 1G and 2G ECs. Characterizing EC puff topographies is necessary for establishing standardized vaping machine protocols [16] which accurately reflect actual product use behavior for the distinct generation of device. This is needed to fulfill the testing requirements of the U.S. food and drug administration (FDA) for premarket tobacco product applications (PMTA). PMTA guidelines instruct applicants to assess the chemical profile of their device when used with a variety of accompanying products under normal and intense use conditions. Since many of these products are new and represent an incremental step along the current device trends, the normal and intense use conditions are unknown and are likely different from older EC products. Floyd et al. [29] studied the effect of flow rate on nicotine yield at different power settings (25–75 watts) using a 3G, sub-ohm EC. They compared the Cooperation Centre for Scientific Research Relative to Tobacco (CORESTA) guideline No 81 recommended flow rate of 1.1 L/min to three higher flow rates up to 6.0 L/min. Key findings showed about 2.5 times greater nicotine yield as the flow rate was increased. This further shows the need for realistic EC flow rates when conducting laboratory evaluations with simulated puffing. In order to simulate realistic puff flow rates, we must have puff topography devices capable of measuring these higher flow rates.

Current smoking topography devices are relatively small, mobile, and capable of data logging, thus allowing their use in laboratory or field settings. These devices measure and store data in real time which can then be extracted afterward [30]. Cigarette smoking topography devices have often been adapted and employed in the assessment of EC puff topography [16,31] with significant differences observed between smoking and vaping topographies [10,32,33]. A typical difference observed in early EC topography studies was longer EC puff duration, even though cotinine levels were lower in EC users [10,34]. However, more recent topography studies that included newer, higher-powered devices have observed similar puff durations, but higher puff flow rates [29] and puff volumes [15] which may be attributed to the direct lung inhalation techniques commonly used for these devices.

Unlike cigarette smoking topography, which has been widely researched [35], there is a paucity of information about EC topography [24,36], which is concerning considering the extraordinary variety in devices across the generations. The assortment of ECs compared to traditional cigarettes makes it difficult to assess puff parameters across users, parameters such as number of puffs, puff volumes, and puff duration. Some studies have elected to use self-reported number of puffs to assess EC topography, however this form of topography is very limited in its scope since subjects tend to under report usage [37] and may not accurately recall the number of puffs taken [38]. Puff volumes also vary widely across EC topography studies [16,34,39] and have ranged between 50–600 mL. Furthermore, ECs tend to be used over a wide range of puff sessions (1–100 puff sessions) whereas traditional cigarette smoking sessions tend to correspond with whole cigarettes smoked and are limited to intervals of 10–12 puffs. This makes it difficult to quantify EC use compared to traditional cigarette use which can be easily assessed based on number of cigarettes consumed during smoking sessions [38,40].

There are a few devices commonly used to evaluate EC topography. The Clinical Research Support System for Laboratories (CReSS) pocket and Smoking Puff Analyzer (SPA-D) are commercially available devices that were compared by Mikheev et al. [41] while collecting EC puff topography. Performance at higher flow rates varied between the two devices with the CReSS device consistently underperforming, which was concluded to be a limitation of that device. Another topography device, Wireless Personal Use Monitor (wPUM), was developed by Robinson et al. [42]. This device was designed and developed for topography measurements across the various emerging tobacco products but is not widely available. Currently, there are few known topography devices designed solely for the purpose of gathering EC puff topography information, and none especially made for sub-ohm, third generation (3G) devices even though these devices are obviously puffed with larger flow rates and volumes. Kong et al. analyzed YouTube videos showing extreme examples of 3G EC use [43] in cloud blowing competitions and vape trick montages. In these videos, vapers are shown taking lung volume puffs in about 2.5 s. This implies a puff flow rate of about 75 L/min, which is far greater than the typical puff rates of cigarettes around 1–2 L/min. The purpose of the present study was to develop and validate a puff topography system capable of measuring the puff topography of sub-ohm, 3G devices as used by vapers in a naturalistic setting such as a vape shop.

## 2. Materials and Methods

We built a differential pressure flow cell based on Bernoulli equation that can measure very high flow rates (0–70 L/min) and that imparted minimal flow resistance (~2%) so that the flow cell does not perceptibly alter the EC user’s experience (Figure 1). We accomplished this by designing around the limiting geometries of several EC atomizers; in particular, we focused on the internal flow channel that passes the heating coils and wick. We used computer aided design and three-dimensional (3D) printing to rapidly construct functional prototypes, then performed flow calibration and volume challenges using primary standards.

### 2.1. Bernoulli Flow Cell Design Description

As a fluid passes through an expansion or constriction, the velocity of the fluid changes, which alters the kinetic energy of the fluid. The change in kinetic energy is balanced by a corresponding increase or decrease in potential energy in the form of pressure; this is known as the Venturi effect. The pressure change is proportional to the difference in the square of velocities of the fluid as it passes through the expansion or constriction [44]. A 3D model of a flow cell was designed in TinkerCAD, a free, web-based computer aided drafting program to utilize the Venturi effect. The 3D model was prototyped by 3D printing with an AnyCubic Photon (AnyCubic, Shenzhen, China) LCD-based Stereolithography printer which uses 405 nm photosensitive resin. Layer thickness was set to 50 μm with 12 s of cure time per layer, and 30 s cure time for the four base layers. Printed flow cells were cleaned with 91% isopropyl alcohol. Pressure sensors and data logger were mounted in a housing with tubing connecting to the flow cell. The flow cell was designed to first expand the fluid (22 mm diameter), causing pressure increase, then to constrict the fluid to the same internal diameter of a typical 3G electronic cigarette mouthpiece (6.5 mm diameter), causing pressure decrease. The overall expansion ratio was 3.4. Pressure measurement ports were integrated into the 3D design and placed at the expansion and constriction points of the flow cell, as shown in Figure 1A. Pressure measurement was conducted by external pressure sensors (Amplified Low-Pressure Sensor, All Sensors, Morgan Hill, CA, USA; purchased through Mouser.com) connected to the flow cell by flexible tubing.

The flow cell was designed to fit between the user’s atomizer and mouthpiece. Atomizers with an “810” mouthpiece were a direct fit and atomizers with a “510” mouthpiece required an adaptor that we designed. An “810” mouthpiece fits a socket of ~12.5 mm diameter and 11 mm depth. The 810 mouthpiece was selected after consultation with several local vape shops regarding the most common mouthpiece sizes in our local vaping community. Ten replicate flow cells were 3D printed and evaluated as described below to characterize the reproducibility of our 3D print fabrication technique.

### 2.2. Data Acquisition System

A Dataq DI-245 data acquisition system (DATAQ Instruments, Akron, OH, USA) was used to collect sensor data. Data acquisition software (WinDAQ, DATAQ Instruments, Akron, OH, USA) was provided with the DI-245 unit by the manufacturer and used in this study. The sampling rate was set to 25 hz for each sensor, although much higher sampling rates were possible.

### 2.3. Flow Cell Evaluation

Once printed and cleaned, all flow cells were evaluated for their relationship between pressure drop and flow rate to determine the variability in our production process. This was performed by drawing a known flow rate through the flow cell from 0–70 L/min. A challenge flow was generated using a scroll type vacuum pump, metering valve, and verified with a primary standard flow meter with +/− 1% accuracy (Defender 520, Mesa Labs, Lakewood, CO, USA). The pump flow was pass through two separate 1L vessels to act as pulsation dampeners prior to connection to the flow cell mouthpiece. For initial validation, pressure differential measurements were collected for each of the 10 replicate flow cells using a digital manometer (DP-Calc Micromanometer 5825, TSI, Shoreview, MN, USA). The manometer was set to 1 s averaging and measurements were read from the screen and recorded manually. A best fit regression for all combined flow cell measurements was used to form a prediction equation representing all flow cells. Using a statistical analysis software (SAS version 9.4), a 95% confidence limit of the regressions was calculated to show the reproducibility of our flow cell fabrication process. The relationship between pressure and flow rate was expected to follow a second order (quadratic) relationship based off the Bernoulli equation solution; therefore, modeling began with this theoretical basis. This means that a linear relationship was expected between the square root of pressure and flow rate. Overall, the relationship was well modeled as linear with R^2^ = 0.998; however, model residuals were systematically biased. Therefore, a quadratic regression of the square root of pressure was used to balance the residuals. Using a quadratic regression of the square root of pressure to model flow also slightly improved the overall fit of the regression to R^2^ = 0.9991.

Since the observed relationship corresponded well with predictions from the Bernoulli equation, we advanced to incorporating a small, high accuracy (+/− 0.25%) pressure sensor into the data acquisition system. Two ranges of pressure sensors were selected: a “low” pressure sensor intended for low flow measurements (0.25 INCH-G-4V, R14C24-17, Amphenol All Sensors, Morgan Hill, CA, USA) and a “high” pressure sensor intended for high flow measurements (5 INCH-D-4V, R15K26-07, Amphenol All Sensors, Morgan Hill, CA, USA). These pressure sensors were attached to a randomly selected flow cell and the relationship between pressure sensor voltage (derived from pressure measurement) and flow rate was determined for both sensors across their full measurement range in the same manner as described above. Establishing this relationship calibrates the sensor output to flow through the flow cell as incorporated into the system as a whole.

After calibration of the flow cell with pressure sensors, many known-volume challenge puffs were generated with a 3-L spirometry calibration syringe (Puritan Bennett VS300 3L Calibration syringe). Challenge puffs were generated manually to a known volume stop point set on the calibration syringe plunger. Challenge puffs were conducted slowly to simulate lower flow rate puffing and quickly to simulate high flow rate puffing. Three puff volumes were conducted at slow and fast puff rates, 0.71, 2.00, and 3.00 L. Each challenge puff condition was simulated 10–15 times and details of the puff simulation are as follows.

For 0.71 L puffs, slow puffs were conducted across 3.0 to 4.5 s (9.5–14.2 L/min) and fast puffs were conducted across 1.1 to 2.0 s (21.3–38.7 L/min). For 2.00 L puffs, slow puffs were conducted across 7 to 11 s (10.9–17.1 L/min) and fast puffs were conducted across 2.1 to 2.5 s (48.0–57.1 L/min). For 3.00 L puffs, slow puffs were conducted across 11 to 14 s (12.9–16.4 L/min) and fast puffs were conducted across 1.8 to 5.0 s (36–100 L/min). Pressure sensor voltage was converted into flow rate using the calibration equation. Flow rate was integrated across puff duration using a simple Riemann sums approach to estimate the volume of these known-volume puffs. These simulated puffs were used to evaluate the robustness of the multi range pressure sensor array with the expectation that the low-pressure sensor would provide more accurate estimation of smaller volume, low flow puffs while the high-pressure sensor would provide more accurate estimation of the larger volume, high flow puffs.

### 2.4. Flow Cell Cleaning, Maintenance, and Durability

Flow cells should be cleaned and sanitized using 91% isopropyl alcohol solution when used with human subjects. The outside of the flow cell and tubing should be wiped or sprayed with alcohol and the interior should be rinsed with 1–2 mL of alcohol from a squirt bottle after each use. Care should be taken to not obstruct the pressure ports with liquid alcohol during rinsing; however, due to the relatively large diameter (1.8 mm) of the pressure ports and low surface tension of isopropyl alcohol, this was not an issue during a series of puff simulations conducted with nicotine containing e-liquid. The flow cell should be allowed to fully dry before the next use which can be accelerated by passing clean air through the flow cell after sanitizing. During use, the pressure ports should be maintained upward to prevent pooling of e-cigarette aerosol that deposits within the flow cell. However, this was not an issue during our test puffs above. It is possible that a prolonged vaping session could result in concerning levels of buildup within the flow cell. If the pressure ports or tubing do become clogged with liquid, the system will cease to measure pressure differential correctly. This can be resolved by disconnecting the tubing and blowing dry air through the tubing towards the flow cell and reconnecting the tubing. Other than regular cleaning, we expect very little maintenance is necessary for these flow cells.

### 2.5. Data Analysis

All statistical analysis was performed using SAS version 9.4 (SAS, Cary, NC, USA), while data reduction and calculation of puff flow rates and puff volumes was performed first in Microsoft Excel and then incorporated into a self-coded Python program based on the regression model from SAS. No automated puff identification was developed for these initial validation tests. When this system is used to collect data from human subjects, those data will be used to create a set of puff identification parameters.

## 3. Results

### 3.1. Flow Rate—Pressure Model

As noted in the methods section, the relationship between flow rate and pressure differentials was expected to follow a square root relationship. Due to minor systematic deviations in the residuals plot, we elected to use a quadratic regression (r^2^ = 0.9991; *p* < 0.0001, Figure 2) that balanced the residuals. Note the tight clustering of the individual measurements (black circles) to the regression (solid red line) and the narrow range of the 95% confidence interval (dashed red line) of the regression. This indicated high homogeneity among our 10 replicate flow cells and suitability for using the overall regression instead of a flow-cell specific relationship.

### 3.2. Volume Estimation

Using the two pressure sensors connected to a randomly selected, representative flow cell, the puff volume was estimated for a series of known-volume challenge puffs as described above. Recall that challenge puffs were manually conducted at low and high flow rates at three know volumes to test the capabilities of the two different ranged pressure sensors. The results of known-volume challenge puffs are summarized in Table 1.

For challenge puffs with a known volume of 0.71 L, the mean volume estimation at low flow was 0.73 ± 0.019 L using the low-pressure sensor data, and 0.72 ± 0.011 L using the high-pressure sensor data. Mean volume estimations at high flow rate challenge puffs were 0.43 ± 0.112 L using the low-pressure sensor data, and 0.786 ± 0.011 L using the high-pressure sensor data.

For challenge puffs with known volume of 2.00 L puffed at low flow, the low and high-pressure sensor measurements yielded mean volumes of 1.845 ± 0.028 L and 1.912 ± 0.006 L, respectively. At high flow, the low- and high-pressure sensor measurements yielded mean volumes of 0.744 ± 0.115 L and 2.065 ± 0.012 L, respectively.

For challenge puffs with known volume of 3.00 L, puffed at low flow, the low and high-pressure sensor measurements yielded mean volumes of 2.900 ± 0.021 L and 3.002 ± 0.001 L, respectively. At high flow, the low and high-pressure sensor measurements yielded 1.597 ± 0.057 L and 3.199 ± 0.004 L, respectively.

Taken as a whole, these results demonstrate good accuracy for both the low- and high-pressure sensors at low flow conditions (up to ~15 L/min), but underestimation of flow rate and therefore puff volume by the low-pressure sensor (0.25-inch) when measuring high flow rates (greater than ~15 L/min).

In Figure 3, several representative challenge puffs are shown from the 0.71 L, low flow challenge condition. The pressure profiles measured by the two sensors are very well correlated. The 0.25-inch sensor showed occasional signal blunting during the low flow challenge puffs, as indicted by the arrows, which means the maximum measurement capability of the sensor was reached. The 5.0-inch sensor (red) is far from its signal saturation point of ~5.0 VDC and was able to sense the large spike in pressure seen in the first puff.

In Figure 4, many representative challenge puffs are shown from the 2.00 L, high flow condition. Both sensors are displayed across their full measurement range. The 0.25-inch sensor (blue) saturated instantly for every challenge puff, which is why each peak has a flat top. The 5.0-inch sensor (red) shows many details in the flow profile of the same puff but does not saturate. Much pressure profile detail is lost to the 0.25-inch sensor due to signal saturation which resulted in gross underestimation of the puff volume. This was expected for high flow rate puffs and was the reason we investigated a dual range pressure sensor for this topography system.

## 4. Discussion

Reproducibility between the ten-replicate 3D printed flow cells was very high and resulted in accurate flow rate estimation that was used to make accurate puff volume estimations using the pressure sensor and a data acquisition system described above. The relationship between puff flow rate and square root of pressure differential was not strictly linear as would have been expected from the Bernoulli equation. Model fit using linear regression resulted in systematic bias in the residuals that was balanced by using a quadratic regression. This slight deviation from theory was probably due to non-ideal design of our flow cell since a compact design was prioritized over ideal behavior. However, the quadratic term coefficient is quite small, suggesting very minor deviation from linearity.

Use of dual range pressure sensors was explored in this study to improve the sensitivity of the system for low and high flow rate puff measurements. The 0.25-inch pressure sensor was able to measure pressure with greater precision, but when integrating flow rates across simulated puffs to estimate puff volume, there was no statistical or practical difference in the accuracy of the 0.25-inch sensor versus the 5.0-inch sensor. Moreover, the 0.25-inch pressure sensor was quickly saturated during high flow rate puff simulations, which led us to the conclusion that the overall utility of using dual range pressure sensors was poor for our application. We found that the 5.0-inch pressure sensor performed equally well under high flow and low flow puff conditions as compared to the 0.25-inch sensor which performed poorly at high flow. Therefore, the added complexity of utilizing two different pressure sensors to estimate flow was deemed unnecessary for this topography system.

Volume estimation using a Riemann sums approach of data collected at 25 hertz was accurate within 96.7–103% for challenge volumes from 0.71–3.00 L. The model equation used to convert sensor voltage to flow rate was easily incorporated into a Python code that was able to automatically identify simulated puffs with simple threshold shift in a rolling average of three data points. The Python code developed also performed basic statistical calculations for puff duration, puff volume, puff flow rates, inter puff interval, and number of puffs. The next steps for this research are to package the system into a portable case and fine tune the puff identification coding to detect puffs conducted with real users in naturalistic settings such as vape shops.

## 5. Limitations

This system was designed to be directly compatible with an “810” mouthpiece socket. Most 3G atomizers use 810 mouthpieces, but some use the smaller “510” size which can be accommodated with a simple mouthpiece adaptor insert. Other 3G atomizers may use custom sized mouthpieces which may require a larger “universal” adaptor using a flexible seal. This system was developed, calibrated, and challenged using an 810 mouthpiece. Additionally, this system was developed, calibrated and tested at approximately 300 m elevation. Measurements conducted at substantially different elevations are expected to deviate from our calibrated values. Future iterations of this system will use temperature and ambient pressure correction and flow rates will be output in standard liters per minutes. Not all forms of 3D printing will create a solid, non-porous part that is suitable for this application, such as filament or extrusion-based 3D printing. Some techniques of 3D printing have higher precision but lack durability. The photo-resin used in our application has very high precision (down to 2 μm), but is somewhat brittle. Using another resin with more flexibility would increase the durability of the flow cell when dropped.

## 6. Conclusions

The *typical* and *intense* flow rates of current generation sub-ohm vapers are unknown due to insufficient measurement tools, but these flow rates are requirements of the U.S. Food and Drug Administration’s (FDA) premarket tobacco product application (PMTA). However, the puff topography system described in this paper generates very little resistance to flow, easily fits between a user’s atomizer and mouthpiece, is calibrated to measure flow rates up to 70 L/min, and measures puff volume within 96–103% accuracy. Therefore, we expect our system will be fully capable of measuring *typical* and *intense* flow rates of third generation sub-ohm users. This improvement in topography measurement will allow product developers, PMTA applicants, researchers, and regulators to study how changes in e-cigarette device settings and e-liquid composition affect use behavior and user exposures. The easy-to-use portability of our topography system will allow measurement of puff topography in naturalistic settings such as vape shops instead of controlled laboratory settings. Investigators will be able to conduct puff-playback of naturalistic vaping sessions within a controlled lab setting if they have an adequate puff simulation machine. This will allow high quality estimation of changes in human exposures as a result of changes in device settings and e-liquid ingredients, which is an essential component of the PMTA and allows regulators to make well-informed decisions on product approvals based on representative data.

## Figures and Tables

**Figure 1 ijerph-19-07989-f001:**
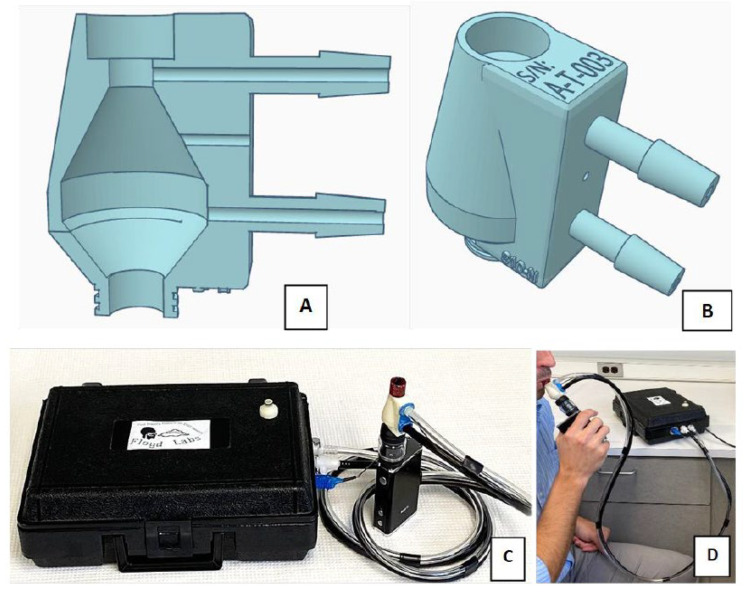
A 3D design of an 810-N puff topography flow cell. Panel (**A**) shows the side profile cut-away of the flow cell with pressure ports located at the constriction and expansion points. Panel (**B**) shows the top socket sized to fit an 810 mouthpiece, and the external view of the device. Panel (**C**) shows the whole topography system connected to an EC device. Panel (**D**) shows the topography device while in use during a simulated puffing session.

**Figure 2 ijerph-19-07989-f002:**
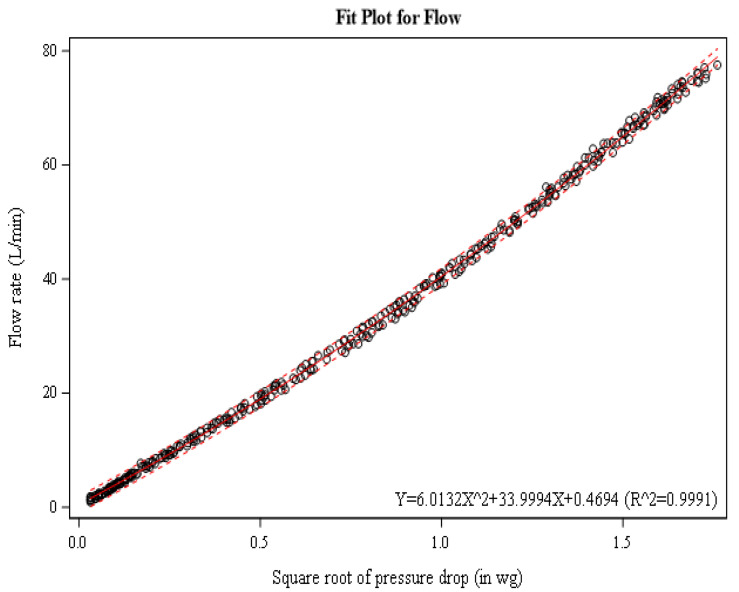
Relationship between square root of pressure drop and flow rate for ten flow cells.

**Figure 3 ijerph-19-07989-f003:**
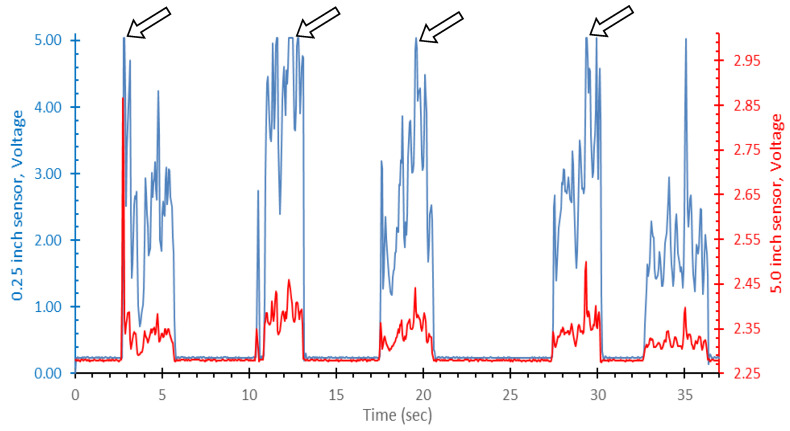
A representative sample plot of sensor voltage (*y*-axis) versus time (*x*-axis) for 0.71 L challenge puffs conducted at low flow. The left-hand scale (blue) corresponds to the 0.25-inch sensor signal. The right-hand scale (red) corresponds to the 5.0-inch sensor signal. The 0.25-inch sensor is displayed across its full measurement range while the 5.0-inch sensor is only displayed across the observed values. Arrows are pointing to instances of signal saturation from the 0.25-inch sensor.

**Figure 4 ijerph-19-07989-f004:**
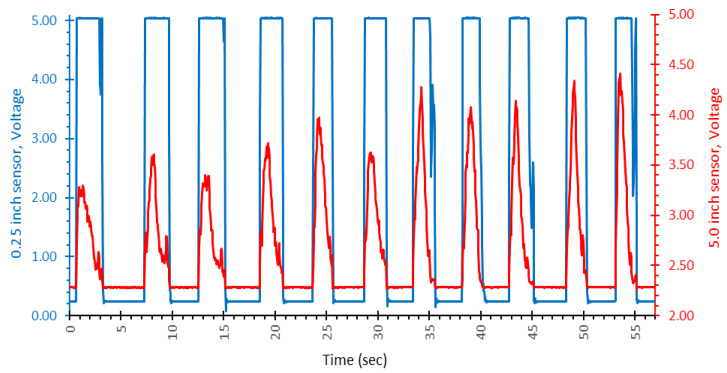
A representative sample plot of sensor voltage (*y*-axis) versus time (*x*-axis) for 2.00 L challenge puffs conducted at high flow. The left-hand scale (blue) corresponds to the 0.25-inch sensor signal. The right-hand scale (red) corresponds to the 5.0-inch sensor signal. Note the pronounced signal saturation in the 0.25-inch sensor as characterized by the flat-topped peak.

**Table 1 ijerph-19-07989-t001:** Volume estimation using topography device with both high- and low-pressure sensors. Simulated puffs conducted manually at low and high flow rates to fixed volume settings on the 3-L calibration syringe.

Challenge Volume	0.710 L Slow		0.710 L Fast		2 L Slow		2 L Fast		3 L Slow		3 L Fast	
Pressure Sensor	Low	High	Low	High	Low	High	Low	High	Low	High	Low	High
Predicted Volume (L)	0.729	0.724	0.431	0.786	1.845	1.912	0.744	2.065	2.900	3.002	1.597	3.199
RSD (%)	0.019	0.011	0.112	0.011	0.028	0.006	0.115	0.012	0.021	0.001	0.057	0.004
Accuracy (%)	103	102	61	111	92	96	37	103	96.7	100.1	53	107

RSD—Relative Standard Deviation expressed in percentage of mean.

## Data Availability

Study data are available upon request from the corresponding author E.F.

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
