# Peer review of "Validation of a High Flow Rate Puff Topography System Designed for Measurement of Sub-Ohm, Third Generation Electronic Nicotine Delivery Systems"

_ijerph, 2022, doi:10.3390/ijerph19137989_

Round 1

Reviewer 1 Report

The paper is very well written and scientifically sound.

However, I have a critical concern, which is is that critical information is missing from the paper that is necessary to replicate it. Without this information, the paper is not particularly interesting because the described device cannot be build or purchased. The principle of the sensor or the prototyping process itself is not very novel and of limited interest.

Preferably, the authors would include the 3D printer files, electronic schematics, PCB layouts, source code, etc as supplementary info.

At the very least, the authors should include the critical dimensions of the 3D printed flow sensor, and describe important design aspects of the electronics in enough detail. For example: Because the pressure transducer (specify part number) is analog, there must have been an analog/digital conversion step. How many bits? With what frequency was the signal sampled? Was any analog or digital signal processing performed on the signal? This kind of information should be in the methods (the sampling frequency was actually  mentioned, but in the discussion).

Reviewer 2 Report

The manuscript titled, “Validation of A High Flow Rate Puff Topography System 2 Designed for Measurement of Sub-Ohm, Third Generation 3 Electronic Nicotine Delivery Systems” is a significant contribution in that the topography device described can be used to fill a significant data gap in the literature regarding puffing behavior of sub-ohm box mod users.  The study developed and tested a 3D printed Bernoulli based topography device that interfaces with the box mode 810 mouthpiece and is capable of measuring/recording flow rates up to 70 L/min.  These data can be analyzed to produce summary topography metrics; i.e., puff volume, puff during, interpuff interval, average puff flow rate, maximum puff flow rate.  Known puff volumes tested were within 96-103% of the true value.  The authors concluded that the device can be used to accurately measure human puffing behavior for sub-ohm box mods, which would fill a critical gap in the literature. The manuscript is well-written and I believe it will be improved with requested changes as outlined below.

The author did a great job in laying out important points in the section ‘What this paper adds’. For future recommendations, the author mentioned about all the products being flavored, and none of them had a tobacco flavor. There are tobacco flavor shisha out in the market so the question here arises is that why there are no tobacco flavored shisha registered in EU-CEG?

Introduction

The introduction focuses to much on the different e-cigarette generations and how the device characteristics differ.  It would be better to focus more on the topography data reported for each of these generations.  I suggest a table showing the four generations of topography devices and the ranges of human puff flow rates, durations and volumes reported for these devices in naturalistic and laboratory settings.  If there are data gaps, they should be highlighted.

Please consider citing the ISO 20768 standard instead of, or at least in addition to, the Tobacco Industry-developed CORESTA standard.

Materials and Methods

There is no information given on the data acquisition system, and the resolution at which the data were acquired.  This should be provided for the pressure sensors and the manometer; if the manometer data were read from a screen and not collected digitally, this should be stated.

There is no discussion of the type of pump that was used, and whether or not its operation involves significant pulsation.  This should be addressed.

There is no discussion on how the flow cell should be cleaned, or if the orientation of the tubing used to make the pressure drop measurements is important to maintaining cleanliness/accurate data. This should be added.

There is no discussion of how the 3D resin that was used will “stand up” to the e-cigarette aerosol over time.  You may not know this at this time, but a theoretical discussion of how the resin and mainstream box mod aerosol will interact with one another should be presented.  For example, will the critical orifices in the flow cell maintain their dimensions over time, and with cleaning?  Also, given that the flow cell was printed in layers, how likely is it that these layers will develop microfractures or separate over time and the flow cell will leak?

Very little detail is provided on how a puff is integrated.  The puffs shown in Figures 3 and 4 are quite noisy.  Is there an algorithm that determines the start and end of each puff, or is this decided on a puff-by-puff basis by a human?  Is any signal smoothing/averaging performed prior to integration? If integrated by a human, then it should be clearly stated that this is not yet a turnkey solution.  If by an algorithm, more detail should be given on mathematically how the start and end of a puff is determined, including any threshold values that are used.

Figure 1 – Panel A and B need the “A” and “B” label.

Change “91% rubbing alcohol” to “91% isopropyl alcohol”.

Page 5, line 203, change “integra” to “integrated”.

Conclusions

The authors describe this as an important tool for PMTAs.  This section should be expanded to include the tool’s utility for MRTPs and Product Standards for e-cigarettes.  For example, understanding how changes in e-liquid ingredients can result in changes to human puffing behavior, and with puff-playback machine smoking, changes in estimated human exposures.

Reviewer 3 Report

The authors describe a new device to measure flow rates and volumes during in situ use of ENDS. The device appears an appropriate contribution to research in ENDS use and the results are convincing. I do suggest the authors provide several additional technical details and considerations to enhance the impact and scientific rigor of their work prior to publication, but otherwise this represents a concise description of instrumentation development. I hope the authors find the following points useful to improve their work:

The motivation for why 3G ECs need to be evaluated is not fully stated. I suggest incorporating a sentence or two to provide this basis. Nicotine yields are mentioned, but there are several other reasons that could be briefly mentioned with relevant works cited.

I commend the authors for clearly defining the distinguishing characteristics of G1-G4 ECs.

Page 2, Line 67: It might be more appropriate to clarify that the standardized protocols should be a set of stand protocols given the wide range of use conditions for ECs.

Page 2, Line 79: You might indicate that it is 2.5 times greater over the range of flow rates listed, not just as flow increases which suggests 2.5 is the maximum.

I suggest Figure 1A be replaced with this same perspective, but showing a cross section to depict the Venturi contained within. Also, A and B are not labeled while C and D are.

Please specify the CAD package used.

I appreciate that the narrow section of the Venturi matches the inner diameter of the EC, but could the authors clarify the expansion ratio they found appropriate and the design considerations that went into this? What is the dead volume of the “manometer” section?

The authors state a few times throughout that pressure drop is negligible (or little resistance to flow), which is a claim I do not doubt, but it should be quantified either theoretically by estimating the coefficient of friction on this material and applying it to the geometry, or by reporting the up and downstream pressures from device inlet to outlet. The inlet to outlet differential is the true pressure drop across the device, but these pressure differentials may well be comparable to those in the “manometer” section (which are indeed quite small); however, it is neither stated nor shown.

The authors may choose to specify that both high- and low-pressure sensors are connected for measurement at all times if this is the case. Further, it would be helpful to readers to earlier standardize the use of 0.25” and 5.0” sensors as low and high pressure.

Given the reliance of Bernoulli’s equation, it may be prudent to include it and explain the assumptions being made. For example, the authors are presumably neglecting the height differential of the device pressure ports. Though, perhaps the equation is not being used in the end (See below).

Overall the manuscript reads well, but there is some puzzling content ordering that lead to misunderstandings in my read through the manuscript. I suggest the authors integrate the entire discussion section into earlier sections of the manuscript as the content discussed explains why the results are presented as such. For example:

In Figure 2, the relationship for square root of the pressure differential vs flow rate, a quadratic fit was employed. The reason for this was very confusing until arriving at the end of the paper where the discussion section clarified why.

Figure 2 it was difficult to discern what the various data points signified. I understand there are 10 devices in use and they apparently behave similarly, but are these all the same symbols. Do they all have points which span the range of pressure drop? What are the three trend lines through the data? 95% confidence interval?

Figure 2 shows deviation from expectation via Bernoulli and it is stated this is due to system non-idealities. Can the authors expand upon what that means? It may also be helpful to show how much it deviates and what the linear fit looks like. I understand that from a practical sense of accurate measurements, one would want to use the trend with better correlation, but there must be an underlying reason for the deviation. It is also not clear how the flow rates on the ordinate were measured. Are these just specified by a mass flow controller, measured by a meter, or both? Is it possible these deviations arise due to this measurement?

The authors have calculated the volume of the puff, presumably through integral analysis with the puff velocities calculated from pressure differential measurement, but …. This was actually clarified later in the discussion section. It is clearer to a reader if this is present in the methods section or when discussing puff volumes. I think the authors mean “Riemann”. How does the 25 hz Riemann discretization compare to the data resolution? Is the resolution so fine that numerical integration of each point is unfeasible? I would suspect higher accuracy with a method that utilizes the full data resolution unless signal noise was problematic.

Table 1 shows an Accuracy %. Is this a mole/volume balance on gas? The accuracy term is a bit vague.

Page 7, Line 252: Was the 0.25 in signal amplified or was the measured output voltage just higher in magnitude compared to the other sensor? Amplified suggests post processing. Further, the scale is larger, suggesting it zoomed out relative to the 5.0 in sensor. This is natural, correct, and not due to post-processing amplification?

Page 7, Line 254: The authors might consider explaining to the readers what the signal blunting indicates in case it is not clear to all readers.

Figure 3, I suggest putting the sensor designation before the word voltage, after a comma, or in parentheses as it seems to say voltage - 0.25 as an operation. Figure 4 is a good example, however this figure has switched from decimals to commas in the ordinate.

I appreciate considerations to the cost of the printer and comments about manufacturing, but it might also be useful to mention the relevance to the reader. In particular, in what capacity will the device be available to others? It is not clear from this manuscript if the device drawings are open source or if the device will be a commercialized product/patent protected. If the later, I suggest declaring competing financial interest for transparency purposes.

Thank you for clarifying the limitations of your calibrations with respect to atmospheric conditions. This was something I was wondering about earlier, but it does fit well in this section. Just something to consider.
